# Identification of Causal Gene-Specific SNP Markers for the Development of Gynoecious Hybrids in Cucumber (*Cucumis sativa* L.) Using the PathoLogic Algorithm

**Manikanda Boopathi Narayanan** [1,*,†] [ID], **Shobhana V. Gnanapanditha Mohan** [1,†], **Praneetha Subramanyam** [2], **Rajasree Venkatachalam** [3] **and Kesavan Markkandan** [4]

1 Department of Plant Biotechnology, Centre for Plant Molecular Biology and Biotechnology, Tamil Nadu Agricultural University, Coimbatore 641003, India
2 Department of Vegetable Crops, Horticultural College and Research Institute, Tamil Nadu Agricultural University, Coimbatore 641003, India
3 Agricultural College and Research Institute, Tamil Nadu Agricultural University, Coimbatore 641003, India
4 ONEOMICS PRIVATE LIMITED, Bharathidasan University Technology Park, Khajamalai Campus, Tiruchirappalli 620023, India
* Correspondence: nmboopathi@tnau.ac.in; Tel.: +91-984-250-9611
† These authors contributed equally to this work.

**Abstract:** Although the genome sequence of cucumber is publicly available, only a limited number of functional markers are in store for developing gynoecious hybrids using Indian genotypes. This study reported novel SNPs and InDels in the exonic regions of genes involved in gynoecy using two parents and their hybrid with genotyping-by-sequencing (GBS) by generating 3.547 Gb of raw data. Using NSDC reference genome GCA_000004075.2, a total of 40,143, 181,008 and 43,612 SNPs were identified, among which 514 were polymorphic between male and female parents but monomorphic between the male parent and the hybrid (confirming hybridity). We further identified that, out of those 514 SNPs, 74 were within the exonic regions of the sex-specific genes. The most interesting functional marker in this study was SNP 2,13,85,488, identified in the gene CsaV3_6G037780 G3I-38214 on chromosome 6, encoding *1-aminocyclopropane-1-carboxylate oxidase 1 (ACS1)*, which plays a key role in female flower production, as indicated in CuCyc with the PathoLogic algorithm. The InDel analysis also identified a variation inside the gene CsaV3_6G304050 G3I-37940, encoding *histone lysine N-methyl transferase*, involved in flowering and female gametophyte development. Thus, this study has identified gynoecy-specific functional markers; upon further validation, these markers will accelerate the evolution of gynoecious hybrids in India and global cucumber breeding programs.

**Keywords:** cucumber; genotype-by-sequencing; gynoecy habit; SNPs/InDels markers





## 1. Introduction

Cucumber (*Cucumis sativus* L.) is a diploid species (2n = 14), belonging to the *Cucurbitaceae* family of the order Cucurbitales, which encompasses 2295 species and 129 genera [1]. Cucumber originated in India [2], particularly in between the Bay of Bengal and the Himalayan Mountain ranges, and it is one of the oldest cultivated crops of the Indian subcontinent. Later, it spread eastward to China [3,4] and westward to Asia Minor, northern tropical Africa and Egypt. Columbus introduced it to Haiti, and later to the USA. Some authors have also reported their origins to be in the tropical African plains and Egypt [3,5].

Both cultivated and wild cucumbers are extensively used in formulating medicines, for corneal disorders to scorpion bites, and it was believed that the fruits could boost fertility in women [6]. It is an affordable and excellent source of vitamins K, B₁ and C; pantothenic acid; and minerals such as phosphorus, magnesium, potassium, manganese, copper, molybdenum and biotin. Cucumber fruits are antioxidative, anti-inflammatory and anti-diabetic in nature [7]. Upon realizing the potential value of cucumbers, they are now

cultivated as an economically and nutritionally important vegetable crop in the subtropical and tropical parts of the world.

Despite the availability of enough literature on its economical and nutritious status and the pan-genomic information of eleven (11) wild and cultivated accessions [8], only minimal genetic and genomic resources are available for cucumber in India. As its primary center of origin, India has the highest diversity ever recorded, but the diversity has gradually depleted due to selection pressure [6]. Owing to this limited genetic diversity and crossability with only a few related species (such as *C. melo*, *C. hystrix* and its wild relative *C. sativus.* var. *hardwickii*), there is no noticeable upsurge in the average yields of cucumbers in the past decades. In order to break the stagnation in yield and enhance the quality, the development of hybrid cucumbers, including gynoecious lines, becomes inevitable.

When monoecious, gynoecious and hermaphrodite phenotypes are present in cucumber [9–13], the gynoecious cultivars can potentially bear more fruits much earlier in the production season. Under greenhouse conditions, gynoecy coupled with parthenocarpy has further enhanced productivity. Gynoecious hybrid varieties of cucumber are predominantly used in the production system of many developed and developing countries, and it is estimated that the annual growth rate of the cucumber hybrid seed market will increase by 4.2% from 2019 to 2027. Asia Pacific is expected to be the major player in a seed market containing hybrid seeds and open pollinated/desi (Indian native types)/heirloom seeds and have a whopping summation of about USD 1939.60 million in 2027 [14]. Thus, hybrid development by using gynoecious lines with regional cucumber cultivars ("desi" types, the popular local/indigenous genotypes of India, meaning "native" types in Hindi) possessing improved qualities would be effective in cucumber breeding programs for exploiting hybrid vigor in yield and quality.

On the other hand, the present day parthenocarpic gynoecious lines are very watery and lack crispiness, resulting in poor quality and market value. However, these lines produce more lateral branches, flowers and fruits and, hence, higher yield. Alternatively, Indian accessions produce crispier, firmer and high-quality fruits, but are poor yielders. Hence, there is a need to combine the qualities of gynoecious cucumbers and desi types by evolving desirable high-yielding hybrids that fetch higher market prices. Nevertheless, understanding the inheritance pattern of gynoecy is imperative to design such hybrid development programs. Research in this direction has provided inconsistent results on the genetic control of female locus (F) in cucumber due to the deployment of diverse sources of gynoecious genes in those studies, as well as environmental effects [15]; consequently, these drawbacks are bottlenecks in conventional breeding programs. Therefore, the breeding of gynoecious lines using molecular markers has been proposed, since selection is performed with genotypes rather than phenotypes [16].

Cucumber has a relatively small genome (367 Mbp). Ever since the first draft of its genome was published [17], several genomic and transcriptomic researches have already been conducted on diversity, biotic and abiotic stress resistances in cucumber [18–26]. Further, the updated version of the genome sequence of 'Chinese Long inbred line 9930' V3 is also publicly available [27]. However, the investigation on elucidating the molecular mechanism of the gynoecy habit in cucumber and integrating such information into the regular hybrid development program is yet to be demonstrated.

The study reported here was designed with the rationale of identifying gene-specific SNP markers that are related to the gynoecy habit. Genotyping-by-sequencing (GBS) was performed using the three cucumber accessions, comprising the gynoecious female parent (Pant-PC2, a popular parthenocarpic variety released from Pantnagar University, Uttarakand, India), the male parent (CBE-CS-33, a regional ecotype collected from Sathur, Tamil Nadu, India) and their hybrid (TNAU PCH1), developed in this university. This study has identified unique SNPs/InDels embedded inside the candidate genes concerning the gynoecy habit. Upon validating these markers in other gynoecious lines possessing such genes, these markers can be employed in commercial hybrid development in cucumbers.

## 2. Materials and Methods

### 2.1. Plant Materials

During the initial period of this study, a total of fifty (50) monoecious land races were collected and screened for their per se performance. Seven (7) land races of cucumber from the fifty land races, namely, CBE-CS-06, CBE-CS-16, CBE-CS-24, CBE-CS- 32, CBE-CS-33, CBE-CS-35 and CBE-CS-36, were collected from different parts of Tamil Nadu, India (Amaravathi, Sempatti, Dharmapuri, Sangagiri, Sathur, Musiri and Kalcheri, respectively). About six (6) tropical gynoecious genotypes (AVCU 1202, AVCU 1203, AVCU 1205, AVCU 1206, AVCU 1302, AVCU 1303 and Pant C2) were further screened for yield and other attributes in order to choose the best parents for hybrid development. The field and polyhouse experiments were conducted at Tamil Nadu Agricultural University, Coimbatore, India, during 2018–2019 and 2019–2020. The tropical gynoecious genotypes were raised inside the polyhouse and the monoecious land races were sown in an open field with three replications. The experiment was laid out following all the recommended agronomic practices, and the genotypes were evaluated based on yield and related traits. These experiments have helped to identify the best male (as they are consistent in yield and quality traits) and gynoecious (which had more numbers of female flowers, leading to higher yields) lines to generate gynoecious hybrids. The crossing programs were started in 2020–2021 in all possible combinations. However, only two hybrid combinations, viz., TNAU PCH 1 (derived from Pant C2 x CBE-CS-33) and TNAU PCH 2 (derived from Pant C2 x CBE-CS-16) had a set sufficient number of fruits. Therefore, only these two hybrids with their parents were sown in the polyhouse during 2021. They were phenotyped for eleven traits (number of primary branches per plant, number of male flowers per plant, number of female flowers per plant, days to first female flower opening, node at which first male and the female flowers opens, fruit length (cm), fruit girth (cm), fruit weight per plant (kg), fruits per plant and yield per plant (kg)). The male and female parents were also self-pollinated and the selfed fruits were tagged and allowed to ripen on the plant itself; seeds were extracted for further use as monecious and gynoecious lines, respectively.

### 2.2. Genotyping-by-Sequencing

Leaf samples (weighing 50–100 mg of fresh weight) were collected from young plants (35–40 days after germination) of female parent, Pant C2 (hereafter referred to as PC2), male parent (CBE-CS-33 (hereafter referred to as Sathur) and their hybrid, TNAU PCH1 (hereafter referred to as PCH1). The tissues were freeze-dried and ground to a fine powder. Genomic DNA was extracted from the samples using the CTAB method and diluted to 80 ng/µL using distilled water. The libraries were constructed following the GBS protocol as described by Elshire [28]. The genomic DNA from individual samples was incubated briefly at 37 °C with *ApeKI* (New England Biolabs (NEB), Ipswich, MA, USA), T4 DNA ligase (NEB), ATP (NEB) and an *ApeKI* Y adapter N-containing barcode. Then, restriction-ligation reactions were heat-inactivated at 65 °C. Next, we performed the PCR reaction using purified samples and PCR products were isolated to retain fragments of approximately 300–350 bp (with indexes and adaptors) from an agarose gel using a Gel Extraction Kit (QIAGEN, Redwood City, CA, USA). The resulting library was sequenced on an Illumina NovaSeq 6000 platform using the 150 bp paired-end protocol. These procedures were performed at ONEOMICS PRIVATE LIMITED, Tiruchirappalli, Tamil Nadu, India (www.oneomics.in), accessed on 3 January 2023.

### 2.3. Reference-Based SNP Calling and Identification of SNPs

We used the in-house pipeline for processing the GBS data owing to their high flexibility and ability to manage complex GBS data sets. Initially, raw GBS reads were de-multiplexed and filtered using the following options: (1) the reads containing the Illumina library construction adapters, (2) the reads with more than 10% of unknown bases (N bases) and (3) one end of the read containing more than 50% of the low-quality bases (sequencing quality value ≤5). Consequently, reads of low quality and reads

with no restriction sites were removed. Subsequently, the clean reads were aligned to the *Cucumis sativus* reference genome NSDC Assembly GCA_000004075.2 using BWA with default parameters and duplicate reads were removed using samtools and PICARD (http://picard.sourceforge.net), accessed on 12 December 2022. The raw SNP/InDel sets were called by samtools by assigning the parameters as '-q 1 -C 50 –m2 -F 0.002 -d 1000'. Finally, the resultant sets were filtered using the following criteria: (1) the mapping quality should be greater than 20, and (2) the depth of the variate position should be greater than four. In addition, ANNOVAR was used for the functional annotation of variants (https://annovar.openbioinformatics.org/en/latest/), accessed on 12 December 2022 and the UCSC known genes were used for annotating genes and regions (http://genome.ucsc.edu/), accessed on 12 December 2022.

### 2.4. Sex-Specific SNP Discovery/Chromosome Strolling

As this study aimed to identify single nucleotide polymorphisms (SNPs) and insertions and deletions (InDels) specifically linked to sex-specific genes in the Cucurbit Genomics Database—CuGenDB (http://cucurbitgenomics.org/), accessed on 20 June 2022, we screened the loci of SNPs and InDels in the male and the female parents and the presence of male specific SNPs and InDels in the hybrid (to confirm its heterozygosity). A locus is sex-linked if it is homozygous in half of the female and heterozygous in at least half of the male samples. The exact base pair location of each male-specific SNP/InDel was located on the Cucumber (Chinese Long) v3 Genome (http://cucyc.feilab.net/CU_V3/), accessed on 20 June 2022, available in CuGenDB and searched along the length of the chromosome for any "direct" or "indirect hit"(by a method called chromosome strolling). Any direct or indirect hits inside a gene concerning reproductive behavior, female or male character or floral traits were documented. "Direct hits" were the SNPs/InDels that were inside the genic/exonic sequences of genes related to female flower production/sex related traits. SNPs/InDels that were found nearby (i.e., within ~1000 bp away from the beginning or from the end of the gene sequence) functional genes related to sex-related/female flower-related genes were referred to as "nearby or indirect hits". The functions of the identified SNPs/InDels, which where already annotated in the pathway section, can be found in CucurbitCyc (http://cucyc.feilab.net), accessed on 20 June 2022 of CuGenDB for the genome of Chinese Long (v3). The Cucyc operates on the Pathway/Genome Database (PGDB), which was generated by the PathoLogic algorithm [29–31], a component of Pathway Tools software version 20.0 (developed by the Bioinformatics Research Group, SRI International, Menlo Park, CA, USA) and MetaCyc version 20.0.

### 2.5. Statistical Analysis

All the data collected from the investigated genotypes were analyzed for standard statistics using IRRISTAT (https://irristat.software.informer.com/), accessed on 20 June 2022.

### 2.6. Data Deposition

The de-multiplexed GBS data used in this study have been deposited in the NCBI Sequence Read Archive (BioProject accession number PRJNA910816; https://www.ncbi.nlm.nih.gov/sra/PRJNA910816), accessed on 20 June 2022.

## 3. Results

### 3.1. Field Evaluation of Parents and Hybrids

In order to select the appropriate male parent, all the investigated regional cucumber accessions were evaluated under open field conditions. Among the seven investigated lines, CBE-CS-32 (3.48 kg/plant), CBE-CS-16 (3.02 kg/plant) and CBE-CS-33 and CBE-CS-06 (3.00 kg/plant) accessions recorded high fruit yield per plant (Table 1). Hence, these three accessions were chosen as the male parents for the development of gynoecious hybrids.

**Table 1.** Growth and yield performance of investigated cucumber lines with three replications.

| Name of the Regional Genotypes | Days to First Female Flower Opening | Number of Male Flowers | Number of Female Flowers | Number of Fruits per Plant | Length of Fruit (cm) | Average Fruit Weight (g) | Yield per Plant (kg) |
|---|---|---|---|---|---|---|---|
| CBE-CS- 06 | 39.80 | 127.00 [D] | 26.60 | 11.00 | 21.12 | 272.27 | 3.00 |
| CBE-CS- 16 | 37.80 | 128.20 [D] | 28.60 | 10.60 | 23.82 | 278.84 | 3.02 |
| CBE-CS- 24 | 37.20 | 126.00 [D] | 30.00 | 8.60 | 19.26 | 253.42 | 2.26 |
| CBE-CS- 32 | 38.40 | 129.40 [D] | 29.80 | 11.20 | 22.02 | 298.80 | 3.48 |
| CBE-CS- 33 | 37.40 | 125.80 [D] | 26.40 | 10.40 | 20.42 | 286.48 | 3.00 |
| CBE-CS- 35 | 40.20 | 120.40 [D] | 29.60 | 9.00 | 12.62 | 215.44 | 1.06 |
| CBE-CS- 36 | 38.40 | 126.80 [D] | 28.00 | 9.40 | 16.68 | 258.68 | 1.54 |
| AVCU 1202 | 33.40 | 9.40 | 43.80 [D] | 5.60 | 14.44 | 250.66 | 1.38 |
| AVCU 1203 | 34.20 | 13.40 | 51.40 [D] | 6.60 | 15.78 | 242.16 | 1.62 |
| AVCU 1205 | 37.60 | 12.60 | 54.20 [D] | 5.40 | 13.70 | 188.34 | 1.00 |
| AVCU 1206 | 30.00 | 14.00 | 64.00 [D] | 8.20 | 17.38 | 225.00 | 1.87 |
| AVCU 1302 | 43.80 | 3.80 | 31.60 [D] | 4.00 | 11.94 | 182.00 | 1.68 |
| AVCU 1303 | 32.20 | 11.99 | 45.40 [D] | 7.60 | 12.04 | 192.66 | 1.50 |
| Pant C2 | 42.80 | 3.20 | 33.50 [D] | 8.00 | 10.40 | 236.00 | 1.82 |
| CD (5%) | 2.87 | 1.56 | 4.47 | 0.63 | 1.87 | 14.98 | 0.73 |

CD (5%) = Critical difference at 5% confidence level; [D]—dominant character.

On the other hand, the results of evaluating the six tropical gynoecious lines under polyhouse conditions exhibited that the gynoecious lines AVCU 1206 (1.87 kg/plant) and Pant C2 (1.82 kg/plant) recorded greater numbers of female flowers and yield (Table 1); hence, they were utilized as female parents for the development of gynoecious hybrids.

Though hybrids were developed by employing all possible cross combinations using the identified male and female parents, only two hybrids, viz., TNAU PCH 1 (derived from Pant C2 x CBE-CS-33) and TNAU PCH 2 (derived from Pant C2 x CBE-CS-16) set fruits. Hence, these two hybrids were sown in the polyhouse and evaluated, along with their parents. Among them, the hybrid TNAU PCH 1 performed better (as it recorded 22.68 fruits/plant with a yield of 3.58 kg/plant in a duration of 132.3 days, whereas TNAU PCH 2 recorded 21.92 fruits/plant with a yield of 3.20 kg/plant in a duration of 139.3 days (Table 2)).

The TNAU PCH 1 has also been identified as the best hybrid when it was compared with the check hybrid KPCH 1. Hence, in order to identify a gynoecious marker for efficient hybrid production, TNAU PCH 1 was selected as experimental material. Leaf samples of a female parent (PC2), male parent (Sathur) and their hybrid (TNAU PCH1) were collected and used for GBS data generation.

**Table 2.** Mean performance of parents and hybrids of cucumber investigated in this study, with three replications.

| S. No | Parents and Hybrids | No. of Primary Branches | No. of Male Flowers/Plant | No. of Female Flowers/Plant | Days to First Female Flower Opening | Node at Which First Male Flower Opens | Node at Which First Female Flower Opens | Fruit Length (cm) | Fruit Girth (cm) | Single Fruit Weight (kg) | No. of Fruits/Plant | Yield/Plant (kg) |
|---|---|---|---|---|---|---|---|---|---|---|---|---|
| | | | | | Male parents | | | | | | | |
| 1. | CBE-CS-33 | 4.78 | 85.29 | 26.42 | 35.46 | 2.56 | 3.65 | 40.85 | 19.42 | 0.141 | 11.29 | 1.591 |
| 2. | CBE-CS-16 | 6.13 | 84.36 | 27.52 | 36.63 | 2.29 | 3.57 | 43.52 | 19.64 | 0.139 | 11.22 | 1.559 |
| | | | | | Female parent | | | | | | | |
| 3. | Pant C 2 | 6.38 | 8.53 | 28.34 | 31.32 | 2.42 | 3.86 | 15.32 | 9.16 | 0.172 | 16.43 | 2.83 |
| | | | | | Hybrids | | | | | | | |
| 4. | TNAU PCH 1 | 6.65 | 5.16 | 31.96 | 23.26 | 2.13 | 3.22 | 16.96 | 9.22 | 0.158 | 22.68 | 3.58 |
| 5. | TNAU PCH 2 | 6.54 | 5.28 | 30.53 | 25.58 | 2.34 | 3.61 | 16.53 | 9.34 | 0.146 | 21.92 | 3.20 |
| | CD | 1.23 | 6.46 | 2.26 | 2.18 | 1.10 | 1.04 | 4.62 | 0.93 | 0.69 | 2.47 | 1.68 |

CD (5%) = Critical difference at 5% confidence level.

### 3.2. Statistics of Genotyping-by-Sequencing (GBS) Data

In total, 3.547 Gb of GBS raw data was generated from these three samples and, subsequently, 3.546 Gb of clean data was generated after filtering low-quality data. The raw data produced for each sample ranged from 1158.117 Mb to 1224.15 Mb, which indicated that a sufficient amount of sequenced data were produced in this study. As the Q30 was in the range of 83.50 to 86.91%, it was also concluded that the sequencing quality could meet the standards for further bioinformatics analyses (Table 3). It was further noticed that the generated GBS data contained 34.84% to 35.8% of GC, which was in the normal distribution range and, hence, fulfilled the quality standard. Thus, it was concluded that the library construction and sequencing procedures were successful and highly reliable. In order to proceed with the physical mapping of these raw reads, the reference genome was downloaded from https://plants.ensembl.org/Cucumis_sativus/Info/Index, accessed on 12 December 2022. The downloaded sequence has 183 sequence numbers with a total length of 193,829,320 bp, 32.37% of GC, 1.8% gap rate, 29,076,228 bp N50 length and 19,226,500 bp N90 length.

**Table 3.** Statistics of genotyping-by-sequencing data generated in this study.

| Sample | Raw Reads | Mapped Reads | Q30 (%) | Mapping Rate (%) | Average Depth (X) | GC Content (%) |
|---|---|---|---|---|---|---|
| PCH1 | 7,318,584 | 6,074,701 | 86.91 | 83.00 | 30.80 | 34.84 |
| PC2 | 7,239,920 | 5,954,078 | 84.86 | 82.24 | 32.13 | 35.80 |
| Sathur | 7,515,922 | 3,458,720 | 83.50 | 46.02 | 25.61 | 34.98 |

### 3.3. SNP Detection and Annotation

Using the reference genome and the GBS dataset, a total of 40,143, 181,008 and 43,612 SNPs were detected in the PC2, Sathur and PCH1, respectively (Figure 1 and Table 4). In order to increase the possibility of detecting unique candidate SNPs that were linked to gynoecy-specific genes, SNPs were only gauged from the exonic regions. In addition, SNPs in the exonic region were further categorized into eight groups, as presented in Table 4.

**Table 4.** Details of SNPs detected among the three investigated lines with respect to the reference genome.

| S. No. | Category | Sub-Category | PC2 | Sathur | PCH1 |
|---|---|---|---|---|---|
| 1 | Non-exonic | Upstream | 3494 | 9950 | 3841 |
| 2 | Non-exonic | Missense | 0 | 0 | 0 |
| 3 | | Stop gain | 25 | 208 | 26 |
| | | Stop loss | 5 | 26 | 4 |
| | Exonic | Synonymous | 876 | 17,107 | 870 |
| | | Non-synonymous | 905 | 13,739 | 890 |
| | | Unknowns | 0 | 0 | 0 |
| 4 | Non-exonic | Intronic | 5674 | 65,725 | 5991 |
| 5 | Non-exonic | Splicing | 15 | 93 | 10 |
| 6 | Non-exonic | Downstream | 3397 | 10,134 | 3705 |
| 7 | Non-exonic | Upstream/downstream | 516 | 1511 | 583 |
| 8 | Non-exonic | Intergenic | 24,344 | 50,281 | 26,815 |
| 9 | Non-exonic | ts | 25,208 | 109,443 | 27,432 |
| 10 | Non-exonic | tv | 14,935 | 71,565 | 16,180 |
| 11 | Non-exonic | ts/tv | 1.687 | 1.529 | 1.695 |

**Table 4.** *Cont.*

| S. No. | Category | Sub-Category | PC2 | Sathur | PCH1 |
|---|---|---|---|---|---|
| 12 | Non-exonic | Het rate | 0.029 | 0.096 | 0.024 |
| | | Total | 40,143 | 181,008 | 43,612 |

**Upstream:** SNPs located within 1 kb upstream (away from transcription start site) of the gene. **Exonic:** SNPs located in exonic region; non-synonymous: single nucleotide mutation with changing amino acid sequence; stop gain/loss: a nonsynonymous SNP that leads to the introduction/removal of stop codon at the variant site; synonymous: single nucleotide mutation without changing amino acid sequence. **Intronic:** SNPs located in intronic region. **Splicing:** SNPs located in the splicing site (2 bp range of the intron/exon boundary). **Downstream:** SNPs located within 1 kb downstream (away from transcription termination site) of the gene region. **Upstream/downstream:** SNPs located within the <2 kb intergenic region, which is in 1 kb downstream or upstream of the genes. **Intergenic:** SNPs located within the >2 kb intergenic region. **ts:** transitions, a point mutation that changes a purine nucleotide to another purine (A ↔ G) or a pyrimidine nucleotide to another pyrimidine (C ↔ T). Approximately two out of three SNPs are transitions. **tv:** transversions, the substitution of a (two ring) purine for a (one ring) pyrimidine or vice versa. **ts/tv:** the ratio of transitions to transversions. **Het rate:** genome-wide heterozygous rate, calculated by the ratio of heterozygous SNPs to the total number of genome bases. **Total:** the total number of SNPs.

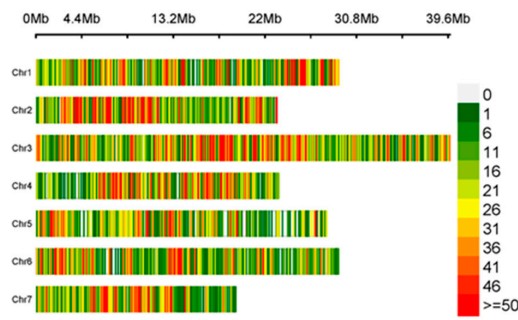

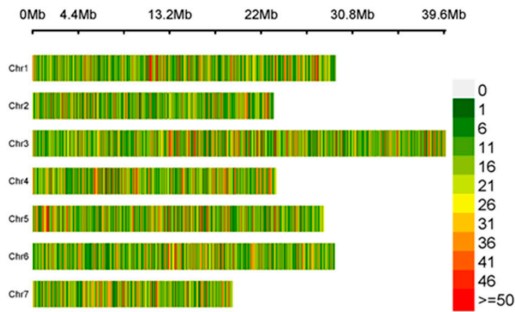

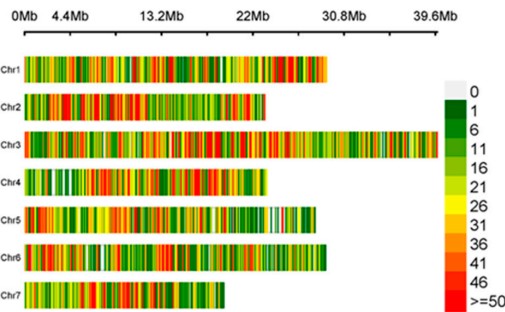

**Figure 1.** Graphical representation of SNPs identified in this study using the *Cucumis sativus* reference genome NSDC Assembly GCA_000004075.2.

A total of 514 SNPs were found to be polymorphic between male (Sathur) and female (PC 2) parents, but monomorphic between the male parent (Sathur) and the hybrid (PCH 1). Out of the 514 SNPs identified as matching between male parent and hybrid (and, thus, they confirmed the hybridity), 74 SNPs were found within the exonic regions of the sex-specific genes as annotated in the reference genome (Supplementary Tables S1 and S2). The exact base pair positions of these 74 male-specific SNPs were located on the Cucumber (Chinese Long) v3 Genome [27]—(http://cucyc.feilab.net/CU_V3/), accessed on 20 June 2022—which is available in the Cucurbit Genomics Database—CuGenDB (http://cucurbitgenomics.org/), accessed on 20 June 2022 [32]. The total number of direct functional hits for each male-specific SNP across the Cucurbit Genomics Database was twenty-five. Chromosome 1 had nine hits; chromosome 2 had two hits; chromosome 3 had three hits; chromosome 5 had a single hit; chromosome 6 had six hits; chromosome 7 had four hits; and there was no hit found on chromosome 4 (Supplementary Table S2).

In the search for SNPs that were present inside any exonic region of a gene, which control the trait/character related to reproduction/flowering/fruiting behavior/female flowers, we attained nine direct hits (Table 5). The transcription factor BIM 1, the xyloglucan endotransglucosylate/hydrolase, the receptor-like kinase (chromosome 1), the zinc knuckle CCHC type family protein, the cyclin dependent kinase inhibitor (chromosome 3), the 6-Phosphofructo-2 kinase/fructose 2, 6-bisphosphate, the pectinesterase, the vacuolar protein sorting associated protein (chromosome 6) and a serine type peptidase (chromosome 7) were the direct gene hits. A major finding here was the SNP 21,385,488 nearby the gene *CsaV3_6G037780 G3I-38214* on chromosome 6 that encodes *1-aminocyclopropane-1-carboxylate oxidase 1 (ACS1)*, which is directly involved in ethylene biosynthesis in plants, a key player in female flower production.

**Table 5.** Number of functional hits for the male-allele-specific SNPs across the Cucurbit Genomics Database.

| Chromosome | | Direct Functional Gene Hits Related to Female Flowering/Fruiting Traits |
|:---:|:---|:---|
| 1 | 1. <br> 2. <br> 3. | Transcription factor BIM 1 <br> Xyloglucan endotransglucosylate/hydrolase <br> Receptor-like kinase |
| 3 | 4. | Zinc knuckle CCHC type family protein |
| 6 | 5. <br> 6. <br> 7. <br> 8. | ***1-aminocyclopropane-1-carboxylate oxidase 1 (ACS1)*** * <br> 6-Phosphofructo-2 kinase/fructose 2, 6-bisphosphate <br> Pectinesterase <br> Vacuolar protein sorting associated protein |
| 7 | 9. <br> 10. | Vacuolar sorting associated protein 2 <br> Serine type peptidase |

* *ACS1* gene that was an indirect hit of SNP 21385488 (400 bp away).

Consequently, the list of reference genome-specific *ACS1* and *ACS2* genes that were deposited in NCBI database were retrieved (Supplementary Table S3), and male-parent-specific SNPs that were located near those *ACS1* and *ACS2* genes were also identified (Table 6).

### 3.4. InDel Detection and Annotation

During the course of InDel detection in this dataset, the female parent, male parent and the hybrid accounted for a sum of 4735, 6218 and 5246 InDels, respectively (Table 7). The male-allele-specific InDels that were found in the hybrid were searched for their presence in the exonic regions, exclusively for their candidacy in female flower expression. Under this restriction, only 56 and 312 InDels present in the hybrid and the male parent were searched for matches, respectively.

Consequently, four InDels were found to be the same in both the male parent and the hybrid. Among them, three were found near the exonic locus and one InDel was found inside the gene *CsaV3_6G304050 G3I-37940*, encoding histone lysine N-methyl transferase with the functions of zinc ion binding and transferase activity protein methylation on chromosome 6 (Table 8).

**Table 6.** List of SNPs identified in the vicinity of *ACS1* and *ACS2* genes that were retrieved from the NCBI Database.

| Chr | Position | Reference | Allele | Gene | Annotated Position | Female | Male | Hybrid |
|---|---|---|---|---|---|---|---|---|
| 4 | 17433658 | C | C T | gene:Csa_4G499310,gene:Csa_4G499320 | intergenic | C C | T T | T T |
| 4 | 17433667 | A | A C | gene:Csa_4G499310,gene:Csa_4G499320 | intergenic | A A | C C | C C |
| 4 | 17433668 | T | T C | gene:Csa_4G499310,gene:Csa_4G499320 | intergenic | T T | C C | C C |
| 4 | 17433722 | G | G A | gene:Csa_4G499310,gene:Csa_4G499320 | intergenic | G G | A A | A A |
| 4 | 17433723 | A | A C | gene:Csa_4G499310,gene:Csa_4G499320 | intergenic | A A | C C | C C |
| 4 | 17433727 | G | G C | gene:Csa_4G499310,gene:Csa_4G499320 | intergenic | G G | C C | C C |
| 4 | 17433812 | A | A G | gene:Csa_4G499310,gene:Csa_4G499320 | intergenic | A A | G G | G G |
| 4 | 17433857 | A | A G | gene:Csa_4G499310,gene:Csa_4G499320 | intergenic | A A | G G | G G |
| 6 | 11413220 | T | T C | gene:Csa_6G168270,gene:Csa_6G169270 | intergenic | T C | T C | T C |
| 6 | 11413245 | G | G A | gene:Csa_6G168270,gene:Csa_6G169270 | intergenic | G A | G G | G G |
| 6 | 11413259 | T | T A | gene:Csa_6G168270,gene:Csa_6G169270 | intergenic | T A | T T | T T |
| 6 | 11413283 | T | T C | gene:Csa_6G168270,gene:Csa_6G169270 | intergenic | T T | T C | T C |
| 6 | 11413317 | A | A G | gene:Csa_6G168270,gene:Csa_6G169270 | intergenic | A G | A G | A G |
| 6 | 11907031 | A | A G | gene | nonsynonymous SNV | A G | A A | A A |
| 6 | 29069874 | T | T C | gene:Csa_6G538800,NONE | intergenic | T C | T T | T T |
| 6 | 29069890 | A | A G | gene:Csa_6G538800,NONE | intergenic | A G | A A | A A |

**Table 7.** Statistics of InDel detection and annotation. Descriptions of the categories are as that of Table 4.

| S. No. | | Category | PC2 | Sathur | PCH1 |
|---|---|---|---|---|---|
| 1. | | Upstream | 583 | 623 | 652 |
| 2. | | Stop gain | 0 | 4 | 0 |
| 3. | | Stop loss | 0 | 1 | 0 |
| 4. | Exonic | Frameshift deletion | 10 | 47 | 9 |
| 5. | | Frameshift insertion | 11 | 53 | 11 |
| 6. | | Non-frameshift deletion | 17 | 102 | 18 |
| 7. | | Non-frameshift insertion | 17 | 104 | 17 |
| 8. | | Intronic | 805 | 2616 | 850 |
| 9. | | Splicing | 2 | 14 | 4 |
| 10. | | Downstream | 544 | 501 | 610 |
| 11. | | Upstream/downstream | 81 | 80 | 92 |
| 12. | | Intergenic | 2459 | 1275 | 2760 |
| 13. | | Insertion | 2361 | 3115 | 2620 |
| 14. | | Deletion | 2374 | 3103 | 2626 |
| | | Het Rate | 0.002 | 0.001 | 0.001 |
| | | Total | 4735 | 6218 | 5246 |

**Table 8.** List of the male-specific InDels identified on the genes involved in flowering.

| Chr | Position 1 | Position 2 | Reference | Altered | Genotype | Annotated Position | Gene ID | Function | Location | Hit |
|-----|-----------|-----------|-----------|---------|----------|--------------------|---------|----------|----------|-----|
| 1 | 2,668,764 | 2,668,764 | - | GT | Homozygous | Frameshift insertion | *CsaV3_1G004260 G3I-29448* | Mitogen activated protein kinase–Cell proliferation, hormonal signaling | 2,662,730–2,668,666 | Nearby |
| 2 | 4,929,433 | 4,929,441 | TTAGTAGTA | AAG | Homozygous | Non frameshift deletion | *CsaV3_2G008570 G3I-43185* | Villin protein–structural protein in microvilli | 4,915,559–4,927,778 | Nearby |
| | 4,929,451 | 4,929,456 | GTAGTA | - | Homozygous | | | | | |
| 3 | 33,422,942 | 33,422,942 | - | AAG | Homozygous | Non frameshift insertion | *CsaV3_3G040770 G3I-28088* | Nucleotide sugar transporter family protein–glycosyltransferases substrates | 33,416,748–33,422,008 | Nearby |
| 6 | 18,785,551 | 18,785,552 | TC | - | Homozygous | Frameshift deletion | *CsaV3_6G304050 G3I-37940* | Histone lysine N-methyl transferase–Zinc ion binding, transferase activity protein methylation | 18,865,436–18,887,620 | Direct |
| | 18,885,449 | 18,885,449 | - | TC | | Frameshift insertion | | | | |
| | 18,885,492 | 18,885,501 | CATTCTCCAT | - | | Frameshift deletion | | | | |

## 4. Discussion

Advanced molecular breeding strategies are promising in improving the production per area, as well as improving quality traits of major vegetable crops in a shorter period. The development of gynoecious hybrids in cucumber has enormous importance for its successful cultivation under both open and polyhouse conditions because of the time and cost incurred in hybrid seed production [33]. Though the present-day gynoecious lines in India (which dominantly produce female flowers) are high yielders, the fruit quality is very low and unmarketable. On the other hand, the regional Indian types produce fruits with more male flowers and high quality, but they are poor yielders (Table 1). Thus, introgressing the best of both the lines into gynoecious hybrids would be an effective and affordable strategy to increase yield and quality simultaneously.

The yield of cucumber fruits is chiefly determined by the ratio of female and male flowers. Sex expression in cucurbits is influenced by genetic, environmental and hormonal factors [34]. Monoecious strains of cucumber bear staminate (male) and pistillate (female) flowers. Gynoecious strains normally produce pistillate flowers only. Ethylene is highly correlated with the femaleness by inducing female flowers and has been regarded as the primary sex determination factor [35,36] and the molecular mechanism behind its involvement has been well-documented [37–45]. Earlier genetics studies indicated that there are three major sex-determining genes in cucumber and melon: *F*, *A* and *M*. Recently, the *A* gene in melon and the *M* gene in cucumber have been cloned and both encode *1-aminocyclopropane-1-carboxylic acid synthase* (*ACS*), which is a key enzyme in ethylene biosynthesis. In cucumber, a series of evidence strongly supports that the *F* gene also encodes *ACS* [46].

So far, four "sex genes" have been identified: *F/CsACS1* [40,41,47], *M/CsACS2* [42,43,48], *A/CsACS11* [49] and *G/CsWIP1* [49,50]. Another gene, *CsACO2*, has also been shown as critical in sex determination by catalyzing the last step of ethylene biosynthesis [50]. Apart from these four "sex genes", *ethylene response 1 (ERT1)*, *ethylene sensitive 3 (EIN3)* and *ethylene responsive factor 110 (ERF110)* were also reported to regulate sex expressions in cucumber [45,51,52]. Yin and Quinn put forth the "one-hormone hypothesis" to emphasize the pressing role of ethylene in the sex expression of cucumber [53]. Besides, gibberellins (GAs) have also shown to have a regulatory role in flower development independent of ethylene [46,54].

Thus, it is imperative to identify functional markers derived from the gynoecious habit, as it would fasten the development of hybrids in cucumber. This study has identified several SNP markers linked to the genes involved in sex determination in cucumber using the algorithm PathoLogic [55]. The genes that are identified in this study have already been functionally annotated and shown to have critical roles in female flower development. In the absence of any experimentally validated markers for the trait of concern, the functionally annotated genes present in publicly available plant databases, such as CuGenDB, serve as valuable genomic resources and a solid base for the initial genic screening of the identified SNPs/InDels across all the chromosomes of the genome.

PathoLogic [55] automatically develops a Pathway/Genome Database (PGDB) describing the metabolic network of an organism. It is the in-built pathway prediction algorithm of the Pathway Tools software suite. Enzymes catalyze metabolic reactions in all organisms, and each enzyme is linked to a reaction. Based on the organization of the reactions and the dataset provided to the algorithm, PathoLogic constructs pathways. It is the simple assumption that experimentally defined metabolic pathways are conserved between organisms; PathoLogic utilizes MetaCyc [56], the non-redundant reference pathway database, as a template for building the metabolic pathways for a newly sequenced organism. PathoLogic has succeeded with 91% accuracy and an F-measure of 0.786. In this way, SNPs/InDels that are identified by GBS are found to be annotated for discrete functions by the PathoLogic algorithm of CucurbitCyc of CuGenDB by the technique of chromosome strolling. These SNPs should be confirmed through functional gene characterization in the follow-up studies.

For example, *Transcription factor BIM 1 (CsaV3_1G003270 G3I-29349)* at 2028622 on chromosome 1 (Supplementary Table S2) encodes a basic helix-loop-helix (*bHLH*) family protein, BIM1 (*BES1-INTERACTING MYC-LIKE 1*), involved in embryonic patterning, brassinosteroid and auxin signaling in Arabidopsis seedlings. They are found in female floral parts such as petals, carpel, petiole, inflorescence and pollens [57]. The *bHLH* family are one of the largest groups of plants TF [58]. They are involved in wound and stress responses [59–62], hormonal regulation [63,64] stigma and anther development and fruit development and differentiation [65–68]. The *bHLH* genes in *Quercus suber* floral libraries have shown to be essential for the development of pollen [69]. Nine groups of *Q. suber* transcripts in female tissues were significantly much expressed, with one transcript (*QsBR ENHANCED EXPRESSION 1*) being exclusive to the female samples [70]. In addition, the *bHLH* gene families are also found to be related to ovule development and regulation of female reproductive development [71]. Therefore, the SNP identified in this study can be used as a potential candidate marker in cucumber hybrid development, at least using the parents investigated in this study.

This study has identified an SNP on *Xyloglucan endotransglucosylate/hydrolase (CsaV3_1 G011020 G3I-30124)* at 6,842,309 bp (Supplementary Table S2), which is associated with primary cell walls as a major tension-bearing structure that limits cell expansion in cauliflower florets [72]. Likewise, SNPs (on *CsaV3_6G042750 G3I-38711* and *CsaV3_6G042750 G3I-38711*; Supplementary Table S2) were also identified in the vacuolar sorting associated proteins (VPSs). They are a division of Endosomal Sorting Complex Required for Transport (ESCRT) engaged in topologically unique membrane bending and scission reactions away from the cytoplasm and VPS4 were found to have roles in centrosome and spindle maintenance in cell division [73]. Liu and co-workers [74] have demonstrated that the absence of VPS38 in plants leads to dampened pollen germination and increased chances of seed abortion.

Similarly, *cyclin-dependent protein kinases (CDKs)* are chief components in cell division and expansion in the cell division processes. They are involved in cell cycle arrest, cyclin-dependent protein serine/threonine inhibitor activity, cell division and expansion, and are directly correlated to the growth of the fruit and the fruit quality. This happens during the very early stages of fruit growth [75]. *CDKs* act as serine threonine kinases in protein complexes along with cyclins in phosphorylating substrates during mitosis. This study identified SNPs in the serine type peptidase gene *CsaV3_7G029970 G3I-47291* (at 18,964,991 bp position on chromosome 7) and the gene code for pectin esterase (*CsaV3_6G038740 G3I-38310*, on chromosome 6 at the 22,176,336th position), which plays major roles in cell division, cell wall formation, cell wall modification and pectin esterase activity. Jiang and his team [75] correlated the upregulation of *CDKs* and *cyclins* with fruit length in cucumber, and other studies have also documented their role in fruit development in cucumber [76] and the early phase of fruit development in avocado [77].

Interestingly, an SNP on the receptor-like kinase 2 (*CsaV3_1G011660 G3I-30188* at the 7,223,169 bp; Supplementary Table S2) that possessed the function of pollen recognition and serine/threonine protein activity [78] was also identified in this study. In the same way, the involvement of 6-phosphofructo-2 kinase/fructose 2, 6-bisphosphate (*CsaV3_6G016080 G3I-37232* on chromosome 6 (involved in phospho–fructose biosynthesis), which was found to have an SNP at 12,041,834 bp in this study) in flower development was also confirmed [79].

Another functional marker that may have vital importance in hybrid development in cucumber is the SNP at 22,706,544 bp on the zinc knuckle CCHC type family protein (*CsaV3_6G026570 G3I-26767*) on chromosome 3, which is involved in DNA binding and zinc ion binding. The zinc-finger family is very diverse in the plant kingdom and consists of a large number of proteins with distinct subfamilies [80]. Proteins containing zinc finger domains regulate diverse signal transduction pathways [81] and abiotic stress responses [79,82–84]. Radcova and his team [85] explored the relationship between the function of the gene coding zinc finger CCHC-type protein and flower morphology and seed size in *Medicago truncatula*. A modified transcript of the gene has resulted in overexpression in anthers. Similarly, an SNP at 1,521,082 on *CsaV3_2G003180 G3I-42754*, which codes for



zinc finger BED-Domain containing protein DAYSLEEPER, is involved in DNA binding and protein dimerization and has been involved in immunity responses in wheat [86]. These genes are found to be an anciently conserved domain in plants, as they were reported in primitive angiosperms [87] and have not been engaged in any of the modern home-keeping genes; hence, they are designated as SLEEPER genes.

Fascinatingly, the InDels reported in this study also have importance in gynoecious hybrid development in cucumber. An example is the InDel at 16,265,469 bp on chromosome 3 of male parent code for the gene *MADS Box transcription factor* (*CsaV3_3G020270 G3I-26344*). The MADS family of transcription factors play significant roles in plant development [70,88] and are key players of flower development in several angiosperms [89]. In another study [90], it was shown that the MADS-box gene encodes a protein similar (85%) to the Short Vegetative Phase (SVP) protein of *Arabidopsis*, which is the popular transcriptional regulator of the flowering time gene.

The significant discovery of this study is the identification of an SNP at 21,385,488 bp on chromosome 6, which was detected 400 bp away from the gene encoding *1-aminocyclopropane-1-carboxylate oxidase 1 (ACS; CsaV3_6G037780 G3I-38214)* (EC Number—1.14.17.4). This gene is directly involved in the ethylene biosynthesis pathway. A single nucleotide change in the ACS gene specifically inhibits the male reproductive organs in melon [91].

Ethylene production is reported to increase drastically during many developmental events such as germination, senescence and abscission, and fruit ripening [92–94]. Ethylene is derived from methionine, which is converted to S- adenosylmethionine by S-adenosylmethionine synthetase (AdoMet). AdoMet is then converted to 1-aminocyclopropane-1-carboxylic acid (ACC) and 5¢-deoxy-5¢methylthioadenosine (MTA) by the 1-aminocyclopropane-1-carboxylase synthase (ACS) [95–97], which is the rate-limiting step in ethylene biosynthesis.

In some transcriptomic studies, the transcript levels of ethylene receptor genes were quantified to be higher in the reproductive organs in plants such as Arabidopsis, roses, rice and tobacco [98,99]. The synthesis of ethylene is related to the sensitivity of ethylene in petals. This is also related to the ACC levels in the flowers. ACC was translocated from the bottom of flowers to the top in the case of roses, orchids and petunia, i.e., from the ovary and receptacle to the pistil, stamen and the lower and upper part of the petals within the flower [99,100]. Ethylene determines the sex of every floral meristem by inhibiting the development of stamens and pistil primordia. The development of stamens or carpels are destined depending on the different concentrations of ethylene, thanks to the varying levels of sensitivity to ethylene in the floral tissues [101–104]. On the contrary, the knock-out-function studies in the orthologous genes, CsACS11 and CmACS11, of cucumber and melon resulted in the complete blocking of the development of female flowers, resulting in androecium [93,105].

The monoecious and dioecious flowers have evolved from bisexual plants either by the arrest of stamens or carpels in a primarily bisexual floral meristem [93,95,97,106–108]. The three major loci, namely, *ACS1/ACS1G*, *ACS11* and *ACO2,* are exclusively involved in the biosynthesis of ethylene at the very early stages of floral meristem development. This regulates the fate of the floral meristem to develop into a female flower [45,86,109]. The homozygous alleles (FF) of the original (*ACS1*) and the duplicate (*ACS1G*) genes produce increased ethylene contents in the floral meristem. This has led to the conversion of monoecious into gynoecious. The heterozygous alleles (Ff) produce decreased levels of ethylene, so the plant turns subgynoecious. The transcript of *CsACS1/CsACS1G* is accumulated in flowers that developed into females, and is present in higher quantities in the shoot apices of gynoecious plants rather than in monoecious ones [47,110,111]

Sex-determining genes were first discovered in cucumber and melon [45,46,50,89]. Many other important discoveries in sex determinations were in pumpkin and squash [100,108,110] and watermelon [112,113], with only slight variations among the mechanisms in all these species [112,113]. Ethylene is the primary regulator of sex determination in cucurbits [95,97,102,111]. External treatments with ethylene are helpful in determining the role of this hormone in the control of sex expression [109–111,114]. Female flowers are

promoted by ethylene by arresting the growth of the stamens or carpels [114,115]. Ethylene also increases the ratio of female to male flowers in *Cucumis* and *Cucurbita* [115,116].

Many factors and stimuli affect the increase in the level of ethylene biosynthesis. Light, wounding, pathogen attacks, biotic and abiotic adversities, hypoxia, toxic chemicals and plant hormones such as auxins, brassinosteroids and cytokinin, and even ethylene itself, can auto-stimulate or auto-inhibit their own rate of production [95,110,117]. The increase in the levels of ethylene during such adverse conditions could be the result of a biological trigger/alarm for the plant to initiate the reproductive cycle at the onset of environmental threats. This is why there is a correlation observed between the biosynthesis of ethylene and female flower production, which should lead to fruiting and seed development for the multiplication of its own species.

At least sixteen (16) different *ACS* and its homologues have been reported in cucumber on chromosomes 2, 3, 4, 5 and 6 (Supplementary Table S3). However, after the previous reports accentuating the close association of gynoecy to the set of *ACS* genes and their homologs in cucumber, we would like to confirm its role in these investigated accessions. Transcriptomic analysis of the same three samples, namely the female and the male parent with the $F_1$ hybrid employed in this study, has clearly indicated that *ACS* was upregulated in the female parent (PC 2) and hybrid (PCH 1), whereas it was downregulated in the male parent (Sathur) (Supplementary Table S4). This clearly depicted the role of *ACS* in the development of female flowers through ethylene biosynthesis, and if this appropriate allele was introgressed into the male parents, it would certainly induce a gynoecious nature. The markers reported in this study (nucleotide variations based on GBS of two parents and their hybrid) are cross-validated across the published reports. Nevertheless, this mere fact may not strongly support any substantial relationship between the markers and the gynoecious phenotype. Furthermore, validation through quantitative trait loci (QTL) mapping or functional genomic analysis in the breeding materials that are generated using the parental lines reported in this study is obligatory.

## 5. Conclusions

The enormous demand for hybrid and desi cucumber varieties keeps heaping up and, thus, developing gynoecious hybrids of cucumber is imperative to keep up with its robust international and domestic markets. Here, we proposed a method of manual screening of SNPs (related to gynoecy) all over the chromosomes of cucumber using CuGenDB. This method may be called "chromosome strolling" since we stroll (walk slowly ) over the length of the chromosome to precisely pinpoint the base pair location of every identified SNP present within a gene or nearby a gene that is annotated for female flower production in the CuGenDB. The CuGenDB is annotated for pathways using an algorithm called PathoLogic, which has automatically constructed metabolic pathways and gene networks rooted in already known information in biological organisms. The SNP at 21,385,488 bp on the chromosome 6, which is 400 bp away from the gene encoding *1-aminocyclopropane-1-carboxylate oxidase 1 (ACS; CsaV3_6G037780 G3I-38214)*, is responsible for ethylene biosynthesis, and therefore in the production of female flowers. This has clearly been shown as a candidate marker that can effectively differentiate male and female parents, but be monomorphic between the male parent and the hybrid, thus confirming hybridity. Hence, hybrid development programs could use these identified candidate markers in the large-scale screening of male parents. Such functional markers would not only help in speeding up the authenticated production of high-yielding gynoecious hybrids, but would also ensure the sustainable and affordable production of cucumber.

**Supplementary Materials:** The following supporting information can be downloaded at: https://www.mdpi.com/article/10.3390/horticulturae9030389/s1, Table S1. Details on male-parent-specific SNPs that were also present in the hybrid; Table S2: List of male-parent-specific SNPs found on the flowering genes; Table S3: List of *ACS1* and *ACS2* genes of Cucumber as described in NCBI Database; Table S4: Identified InDels in the three sequences nearby ACS1 and ACS2 genes found in the NCBI Database; Table S5: List of the male allele specific InDels and the genes related to flowering holding

them; Table S6: Identified SNPs nearby ACS1 and ACS2 genes as listed in NCBI Database; Table S7: Relative expression of flowering specific genes in the investigated lines.

**Author Contributions:** Conceptualization, design of the experiment, writing and reviewing and editing the original draft, M.B.N.; analysis, data curation and writing the original draft, S.V.G.M.; methodology development, fund acquisition and project administration, P.S. and R.V.; processing of the data using in-house pipelines with different bioinformatics software, K.M. All authors have read and agreed to the published version of the manuscript.

**Funding:** Science and Engineering Research Board, Department of Science and Technology, Government of India (Project number: DST/HCRI/CBE/VEG/2018/R003).

**Institutional Review Board Statement:** Not applicable.

**Informed Consent Statement:** Not applicable.

**Data Availability Statement:** Not applicable.

**Conflicts of Interest:** The authors declare no conflict of interest.

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
