# Peer review of "Identification of Causal Gene-Specific SNP Markers for the Development of Gynoecious Hybrids in Cucumber (Cucumis sativa L.) Using the PathoLogic Algorithm"

_horticulturae, doi:10.3390/horticulturae9030389_

Round 1

Reviewer 1 Report

The contents were worth studying and some useful conclusions were obtained. However, there are still some problems in this manuscript. English could be improved. There are some sentences with grammar mistakes and missing articles throughout the manuscript. Some sentences need to be reworded. I would recommend this manuscript after minor revision.

1. Authors claim that experiments were carried out with three replicates. It should be mentioned in tables or figures captions for easy understanding.

2. Literature survey was not enough. Only basic information was provided and that too without the latest references. In particular, gynoecy specific functional markers from other crops and discuss more rather than writing observations.

3. Author should describe a strong conclusion upon their findings.

4. Functional mechanisms of identified SNP is missing in the later part of the discussion. Rather cited single references without proper discussion.  

Author Response

Reviewer 1

The contents were worth studying and some useful conclusions were obtained. However, there are still some problems in this manuscript. English could be improved. There are some sentences with grammar mistakes and missing articles throughout the manuscript. Some sentences need to be reworded. I would recommend this manuscript after minor revision.

1. Authors claim that experiments were carried out with three replicates. It should be mentioned in tables or figures captions for easy understanding.

Details on replication are mentioned in tables and figures.

2. Literature survey was not enough. Only basic information was provided and that too without the latest references. In particular, gynoecy specific functional markers from other crops and discuss more rather than writing observations.

Relevant and recent literature has been included in the discussion.

3. Author should describe a strong conclusion upon their findings.

Conclusion is strengthened.

4. Functional mechanisms of identified SNP is missing in the later part of the discussion. Rather cited single references without proper discussion

Functional details are included.

Can be improved

Does the introduction provide sufficient background and include all relevant references?

(x)

Are all the cited references relevant to the research?

(x)

Is the research design appropriate?

(x)

Are the methods adequately described?

(x)

Are the results clearly presented?

(x)

Are the conclusions supported by the results?

(x)

Reviewer 2 Report

In present study, author identified novel SNPs and InDels in Gynoecious cucumber hybrid using Genotype-By-Sequencing (GBS). Moreover, identified functional SNP marker predicting to   plays key role in female flower production. The presented results are quite interesting while following issues need to address before considering the manuscript.

1.       In abstract, the first line of sentence is not clear, author suggested to improve write-up that reader can easily understand.

2.       Please remove the yellow colour mark from the text of the manuscript.

3.       Introduction is too lengthy, author suggested to concise write-up

4.       In line 75 author said they used regional cucumber cultivars to develop hybrid line, while they did not mention the cultivar name. Author suggested to mention the local cultivar name essential for future research.

5.       In line 148 author said …protocols described in…(Where) please clearly mention it before citing the reference.

6.       If possible author suggested to provide the data about functional study of identified SNPs/InDels for strengthen the key massage of the manuscript.

Author Response

Reviewer 2

In present study, author identified novel SNPs and InDels in Gynoecious cucumber hybrid using Genotype-By-Sequencing (GBS). Moreover, identified functional SNP marker predicting to   plays key role in female flower production. The presented results are quite interesting while following issues need to address before considering the manuscript.

1.       In abstract, the first line of sentence is not clear, author suggested to improve write-up that reader can easily understand.

The first line of abstract is made clear.

2.       Please remove the yellow colour mark from the text of the manuscript.

The colour mark is removed.

3.       Introduction is too lengthy, author suggested to concise write-up

The introduction is made concise.

4.       In line 75 author said they used regional cucumber cultivars to develop hybrid line, while they did not mention the cultivar name. Author suggested to mention the local cultivar name essential for future research.

The details about the parents have been given in lines 104-108.

5.       In line 148 author said …protocols described in…(Where) please clearly mention it before citing the reference.

Reference has been cited appropriately

6.     If possible author suggested to provide the data about functional study of identified SNPs/InDels for strengthen the key massage of the manuscript.

Data provided about the PathoLogic algorithm of CuCyc of CuGenDb

Yes

Can be improved

Must be improved

Not applicable

Does the introduction provide sufficient background and include all relevant references?

( )

( )

(x)

Are all the cited references relevant to the research?

(x)

( )

( )

Is the research design appropriate?

( )

(x)

( )

Are the methods adequately described?

(x)

( )

( )

Are the results clearly presented?

(x)

( )

( )

Are the conclusions supported by the results?

(x)

( )

( )

Reviewer 3 Report

The manuscript entitled Identification of Causal Gene Specific SNP Markers for the Development of Gynoecious Hybrids in Cucumber (Cucumis sativa L.) is well written. Methodology described in details. Identification of SNPs related to  gynoecy has its own importance to develop new cultivars for the purpose of increasing yields.

Theres are some issues, which need to be addressed:

1. keyword need to be revised,  for example, the authors used the word breeding, what kind of breeding have you done in this study?

2. How many genes have been identified by authors related to gynoecy?

3.  Regarding SNP 21385488 how authors believe that this SNP is involved in female flower development without functional characterization.

4. same question for CsaV3_6G037780 G3I-38214 and CsaV3_6G304050 G3I-37940

5. if yes,  which part of  flowers  are influenced by these SNPs  

6.This study lack of sound nevality.How  do authors thinks?

7. I have found some typing errors. Please check through the manuscript.

9. Conclusion should be more focussed on novel identified SNPs associated with flower development rather that cucumber importance

10. Please provide the figure 1 with readable  font size, for example, see Chr1 etc.

Author Response

Reviewer 3

Yes

Can be improved

Must be improved

Does the introduction provide sufficient background and include all relevant references?

( )

(x)

Are all the cited references relevant to the research?

(x)

( )

Is the research design appropriate?

( )

(x)

Are the methods adequately described?

( )

(x)

Are the results clearly presented?

(x)

( )

Are the conclusions supported by the results?

(x)

( )

Reviewer 3

The manuscript entitled Identification of causal gene specific SNP markers for the development of gynoecious hybrids in cucumber (Cucumis sativa L.) is well written. Methodology described in details. Identification of SNPs related to gynoecy has its own importance to develop new cultivars for the purpose of increasing yields.

1. Keyword need to be revised, for example, the authors used the word breeding, what kind of breeding have you done in this study?

The keywords have been revised. The word breeding represents marker assisted breeding which follows up after the identification of the SNP/InDel markers.

2. How many genes have been identified by authors related to gynoecy?

Nine genes are related to gynoecy and have been presented in table 5.

3.  Regarding SNP 21385488 how authors believe that this SNP is involved in female flower development without functional characterization.

The SNP 21385488 is found just 400 bp away from ACS1 gene in chromosome 6 as reported in the CuGenDB. CuGenDb has annotated ACS1 gene for ethylene production through the PathoLogic algorithm. Ethylene production is directly associated with female flower production in cucumber as per previous reports.

4. same question for CsaV3_6G037780 G3I-38214 and CsaV3_6G304050 G3I-37940

CuGenDb has annotated the genes theoretically based on information from metabolic networks of other organisms, which is 91% accurate.

5. if yes,  which part of  flowers  are influenced by these SNPs  

Ethylene acts on stamen arrest and induces carpel development during the early developmental stages. Other SNPs are indirect in their action.

6. This study lack of sound novelty. How do authors think?

The authors have proposed a new method of manual screening of SNPs directly all over the chromosomes of cucumber in CuGenDB. Just like “chromosome walking” this method may be called “chromosome strolling” since we scroll (slowly walk) over the chromosome to precisely detect the base pair location of every identified SNP either within a gene or nearby a gene (within 1000 bp distance) that is annotated for female flower characters in the CuGenDB. As far as we know, there is no such methodology reported in any earlier literature. In this way, the authors think that our manuscript is novel, in a given stance where there is no previous molecular data on SNPs/InDels that are inevitable for MAS in cucumber for gynoecy.  Moreover, the SNPs and InDels identified are associated to the genes involved in gynoecy / female flower development. They are novel since they are reported for the first time

This method will be so much of help for identifying SNPs/InDels in many other orphan crops and perennial/tree crops as well.

7. I have found some typing errors. Please check through the manuscript.

Done.

9. Conclusion should be more focussed on novel identified SNPs associated with flower development rather that cucumber importance

Conclusion revised as suggested.

10. Please provide the figure 1 with readable  font size, for example, see Chr1 etc.

Done.
